# Comparative study of *Plasmodium falciparum msp-1* and *msp-2* Genetic Diversity in Isolates from Rural and Urban Areas in the South of Brazzaville, Republic of Congo

**DOI:** 10.3390/pathogens12050742

**Published:** 2023-05-22

**Authors:** Marcel Tapsou Baina, Abel Lissom, Naura Veil Assioro Doulamo, Jean Claude Djontu, Dieu Merci Umuhoza, Jacques Dollon Mbama-Ntabi, Steve Diafouka-Kietela, Jolivet Mayela, Georges Missontsa, Charles Wondji, Ayola Akim Adegnika, Etienne Nguimbi, Steffen Borrmann, Francine Ntoumi

**Affiliations:** 1Fondation Congolaise Pour la Recherche Médicale, Villa D6-Cité OMS-Djoué, Brazzaville BP69, Brazzaville BP69, Congo; bainamarcel@gmail.com (M.T.B.); assiorodoulamonauraveil@gmail.com (N.V.A.D.); cdjontu@yahoo.fr (J.C.D.); dieumerciumuhoza1@gmail.com (D.M.U.); dollonmbama@gmail.com (J.D.M.-N.); diafkietelas@fcrm-congo.com (S.D.-K.); jolivet.mayela@yahoo.fr (J.M.); georgesmissontsa@fcrm-congo.com (G.M.); etienne.nguimbi@umng.cg (E.N.); 2Département de Biologie Cellulaire et Moléculaire, Faculté des Sciences et Techniques, Université Marien Ngouabi, Brazzaville BP69, Congo; 3Department of Zoology, Faculty of Science, University of Bamenda, Bambili P.O. Box 39, Cameroon; 4Department of Parasitology and Medical Entomology, Centre for Research in Infectious Diseases (CRID), Centre Region, Yaoundé P.O. Box 13501, Cameroon; charles.wondji@crid-cam.net; 5Department of Vector Biology, Liverpool School of Tropical Medicine, Pembroke Place, L3 5QA Liverpool, UK; 6Institute of Tropical Medicine, University Hospital of Tübingen, 72074 Tübingen, Germany; aadegnika@cermel.org (A.A.A.); steffen.borrmann@uni-tuebingen.de (S.B.); 7Centre de Recherches Médicales de Lambaréné, Lambaréné BP242, Gabon; 8German Center for Infection Research (DZIF), partner site Tübingen, 72074 Tübingen, Germany

**Keywords:** *Plasmodium falciparum*, genetic diversity, rural and urban areas, multiplicity of infection, Republic of Congo

## Abstract

Polymorphisms in the genes encoding the merozoite surface proteins msp-1 and msp-2 are widely used markers for characterizing the genetic diversity of Plasmodium falciparum. This study aimed to compare the genetic diversity of circulating parasite strains in rural and urban settings in the Republic of Congo after the introduction of artemisinin-based combination therapy (ACT) in 2006. A cross-sectional survey was conducted from March to September 2021 in rural and urban areas close to Brazzaville, during which *Plasmodium* infection was detected using microscopy (and nested-PCR for submicroscopic infection). The genes coding for *merozoite proteins-1* and *-2* were genotyped by allele-specific nested PCR. Totals of 397 (72.4%) and 151 (27.6%) *P. falciparum* isolates were collected in rural and urban areas, respectively. The K1/msp-1 and FC27/msp-2 allelic families were predominant both in rural (39% and 64%, respectively) and urban (45.4% and 54.5% respectively) areas. The multiplicity of infection (MOI) was higher (*p* = 0.0006) in rural areas (2.9) compared to urban settings (2.4). The rainy season and the positive microscopic infection were associated with an increase in MOI. These findings reveal a higher *P. falciparum* genetic diversity and MOI in the rural setting of the Republic of Congo, which is influenced by the season and the participant clinical status.

## 1. Introduction

Globally, almost half of the population is at risk of *Plasmodium falciparum (P. falciparum)* infection. In 2021, the number of new cases of malaria reached 247 million, causing 619,000 deaths. More than 90% are from sub-Saharan African countries [1]. Poverty, the emergence of insecticide-resistant mosquitoes, the ability of *P. falciparum* to develop resistance to antimalarial drugs, and the unavailability of effective malaria vaccines due to the high genetic diversity within the parasite population are major obstacles to malaria control and elimination [2,3].

In the Republic of Congo, although efforts are devoted to reducing the burden of malaria, it remains one of the major public health problems. Malaria is the leading cause of hospital consultations (63%), hospitalizations (20%), and deaths (9%) [4], with *P. falciparum* species being the leading cause of infection and mortality [5]. In highly malaria-endemic areas, such as the Republic of Congo, *P. falciparum* infection is characterized by high genetic diversity associated with high multiplicity of infection implying the evolutionary fitness of the parasite and therefore a higher survival advantage in overcoming malaria control and elimination efforts [6].

Demographic and entomological data showed that malaria transmission can be reduced via urbanization, including sanitation and education of the population. This increases the number of non-immune individuals, leading to differences in epidemiological characteristics of malaria in rural and urban settings, as many interventions are conducted in urban zones [7]. Interventions on malaria transmission (such as insecticide spraying) might have an impact on the *P. falciparum* strains circulating in an endemic area.

The method commonly used to characterize the genetic diversity of *P. falciparum* isolates in a malaria endemic area is based on the amplification of the genes encoding for the *msp-1* and *msp-2* [8]. These genes are good discriminatory and informative markers for strain differentiation within a parasite population [6,9]. In addition, the study of the genetic diversity of *P. falciparum* and its multiplicity of infection (MOI) describes the intensity of malaria transmission and is a good indicator of the effectiveness of the malaria control and intervention strategy used in a country [10]. A high genetic diversity reported in different regions with necessary control means will then imply that the control program is not effective in reducing the pest population and vice versa. Therefore, information on the rural and urban dynamics of *P. falciparum* genotypes, as well as on the factors determining gene flow between sites, is crucial for successful malaria elimination. However, after the revision of the treatment policy in 2006 in the Republic of Congo, genetic diversity studies have been mainly conducted in the hospital setting and urban areas in order to assess the impact of the implementation of malaria control measures on the circulating parasite strains [11]. This would provide limited information on the circulating *P. falciparum* parasite population in the country. The *msp-1* and *msp-2* genetic polymorphism and the multiplicity of infection in isolates collected from rural and urban setting have been investigated in the Republic of Congo. This work will provide evidence of the effectiveness or not of current malaria control measures in both settings.

## 2. Materials and Methods

### 2.1. Study Sites

The study was conducted in rural areas of the Goma Tsé-Tsé district (Ntoula and Djoumouna villages) in the Department of Pool and in the urban area of Brazzaville Department (Mayanga in the west quarter of Madibou District) of the Republic of Congo. The two departments have a tropical humid climate divided into two seasons: a short period of dry season (June to September) and a long period of rainy season (October to May). These zones are geo-localized at 4.36 S, 15.15 E and an average altitude of 217 m. Djoumouna is a village located 25 km from Brazzaville. It is characterized by the presence of a gallery forest bordering the Djoumouna River. The locality is surrounded by four rivers—the Lomba, Kinkoue, Loumbangala, and Djoumouna—which supply a series of fish-farming ponds [12] that can serve as potential foci of malaria vectors. The population of Djoumouna Village comprises about 800 inhabitants, with agriculture as the main economic activity. Ntoula is a neighboring village of Djoumouna and is irrigated by two rivers. This village accounts around 635 inhabitants mainly living from agriculture and fishing. Mayanga is an urban area located in the 8th Madibou District, to the south of Brazzaville. With a population of about 28,422 inhabitants, Mayanga is characterized by the presence of three sites of market gardening (Agri-Congo 1 and 2 and the Groupement Jean Felicien Mahouna) and is irrigated by three rivers: the Djoué, Laba, and Matou Rivers. Mayanga comprises several public and private services, such as health centers and primary and high schools.

### 2.2. Study Design

A community-based cross-sectional survey was conducted from March to September 2021 covering both the rainy (March to May) and dry seasons (June to September). Individuals at least 1 year old residing in the study areas since at least 3 months prior were included in the study. A well-structured questionnaire was used to record sociodemographic and clinical data, including age, gender, weight, fever or fever history in the last 48 h, headache, vomiting, nausea, etc. Fever was defined based on axillary temperature ≥ 37.5 °C [5]. Using standard aseptic techniques, 4 mL of venous blood was collected in ethylenediaminetetraacetic acid (EDTA) tubes for microscopic examination, hemoglobin determination (using the fully automated CYANHemato cell counter, Cypress Diagnostic, Belgium), and subsequent molecular analysis of isolates.

### 2.3. Microscopic Screening for Plasmodium Infection

*Plasmodium* infection was screened via microscopy using the Lambaréné method [13]. Briefly, 10 µL of venous blood of each sample was distributed on the slide over a rectangular area of 10 × 8 mm to perform a thick smear. The slide was dried at room temperature and stained for 20 min with 20% Giemsa solution (Giemsa R-Solution, Merck, Darmstadt, Germany). Each slide was read by two experienced microscopists. When no agreement was reached on a specific slide, a third microscopist was asked to decide. The number of parasites and the number of high-power fields (HPF) were counted, and the number of parasitemia per microliter was calculated using the microscope factor of 708 as follows: Parasitemia (P/µL) = No. of parasites × Microscope factor/No. HPF.

All participants with symptomatic or asymptomatic parasitemia were treated by medical staff according to World Health Organization (WHO) and National Malaria Control Program treatment protocols.

### 2.4. Molecular Diagnostic of Plasmodium Infection

Parasite genomic DNA was extracted from whole blood using the QIamp DNA kit (Qiagen, Hilden, Germany) following the manufacturer’s instructions. DNA extract samples were then checked for purity and quantity using a nanodrop spectrophotometer. The *Plasmodium* species characterization was performed as we have previously described [5]. Amplified genomic DNA samples confirmed for *P. falciparum* were further characterized for *msp-1* and *msp-2* with specific oligonucleotide primer pairs.

### 2.5. Typing of P. falciparum msp-1 and msp-2 Genes

Nested PCR of the polymorphic regions *msp-1* and *msp-2* (block 2 and 3) was performed as previously described [14] using the primers and amplification conditions reported in Appendix A. Briefly, a primary reaction was performed, amplifying the entirety of block 2 and block 3 of the *msp-1* and *msp-2* genes, respectively. The amplicons of the first reaction were used as templates in six separated secondary PCR reactions using different pairs of primers (Appendix A) that target the specific allelic families of *msp-1* (K1, MAD20 and R033) and *msp-2* (3D7 and FC27). The mix of primary and secondary reactions contained 500 nM of each primer, 2.5 mM MgCl_2_, 100 µM of each dNTP and 0.05 units of Taq DNA polymerase (Qiagen). DNA from *P. falciparum* reference strains (3D7, Dd2 and HB3) was used in each run as positive control. The amplified products were then analyzed via 2.5% ultrapure agarose gel electrophoresis (Agarose UltraPure™ 500 g, Invitrogen, Carlsbad, CA, USA) for *msp-1* and 2% for *msp-2*. The SYBER-Green-stained DNA fragments were visualized using UV transillumination (Gel Doc^TM^ EZ, Biorad) and analyzed with Image Lab Software Version 6.0. Allele sizes were determined with two molecular weight standards (100 bp DNA Ladder, New England Biolabs). Alleles were considered identical if the fragment sizes were within the range of 10 bp [15] and 20 bp [6] for *msp-1* and *msp-2* respectively.

### 2.6. Case Definition

Three groups of *Plasmodium* positive participants were classified in this study:Uncomplicated malaria, also referred as symptomatic malaria—those with positive microscopy and having less than 24 h of fever history and/or symptoms).Asymptomatic malaria was defined as the presence of positive microscopy and no clinical symptoms.Participants presenting no clinical malaria symptoms with a negative microscopy and positive in PCR were classified as sub-microscopic infection.

### 2.7. Definitions of Concepts

Polyclonal infection is defined by the presence of more than one allele for a given gene in a single isolate, whereas the presence of a single allele was considered monoclonal infection. The multiplicity of infection (MOI) is defined as the mean number of distinct parasite clones per infection [16].

### 2.8. Statistical Analyses

Data were recorded in Microsoft Excel (Microsoft Inc., Redmond, WA, USA) version 2016 and analyzed using GraphPad Prism (version 6.0.1). Continuous variables were expressed as means ± standard deviations (SD) or as medians with interquartile range (IQR). The allele frequencies of *msp-1* and *msp-2* were calculated as the proportion of alleles detected for each allelic family out of the total alleles detected. The Mann–Whitney (or Kruskal–Wallis) test was used for comparing non-parametric variables, while Student’s t or analysis of variance (ANOVA) tests were used for parametric variables. χ^2^ tests (or Fisher’s exact test) was used to compare proportions. Statistical significance was defined as *p*-value < 0.05.

### 2.9. Ethical Declaration

This study was approved by the Institutional Ethics Committee of the Congolese Foundation for Medical Research (N°013/CIE/FCRM/2018); administrative authorization was obtained from the Marien Ngouabi University (N°247/UMNG.FST.DFD.FD-SBIO) was obtained, and approval from the head of each study area was also obtained. Written informed consent or assent was obtained from all participants or parents/guardians for those under 18 years of age.

## 3. Results

### 3.1. Characteristics of the Study Population

In total, 548 *Plasmodium falciparum* positive samples were analyzed, including 397 (72.4%) samples from rural and 151 (27.6%) from urban areas. The age of the infected subjects ranged from 1 to 88 years with an average of 14 (1–88) years in rural areas and 16 (3–81) years in urban areas. Participants aged more than 20 years were the primary population both in rural and urban areas. Women were also more represented than men in both study areas. Individuals with positive microscopic infection represented 33.7% (185/548) of the total population, among whom 39.1% (155/397) were from rural areas and 19.9% (30/151) were from urban areas, with geometric means of parasite density of 753 (575–985) parasites/μL and 592 (290–1208) parasites/μL, respectively (Table 1).

### 3.2. Genotyping of P. falciparum msp-1 and msp-2 in the Study Sites

The amplification successes of *msp-1* and *msp-2* genes were 90.2% and 83.9%, respectively (Appendix A). The RO33-type alleles (288/397 (72.5%)) and FC27-type alleles (205/397 (73.3%)) for *msp-1* and *msp-2,* respectively, were found most frequently in isolates collected in rural area, whereas K1-type alleles (104/151 (68.9%)) and 3D7-type alleles (62/151 (41.1%)) were the most detected in isolates from urban areas. The prevalence of the allelic families in general was statistically (*p* < 0.05) different between the two study areas, with the exception of the K1-type family (Table 2). In total, 66 *msp-1* genotypes were identified in *P. falciparum* isolates in rural areas, including (Table 2) 28 K1-type alleles (size range: 120 to 420 bp), 26 MAD20-type alleles (130–410 bp), and 12 RO33-type alleles (110–250 bp) versus 44 detected in urban areas, including 19 K1 alleles (with a size ranging from 150 to 420 bp), 16 MAD20 alleles (120–410 bp) and 9 RO33 alleles (130–210 bp). Overall, the K1 allelic family was predominant—representing 39% and 45.4% of all alleles in rural and urban areas, respectively—followed by the RO33 allelic family (rural: 35% and urban: 32.8%) and MAD20 (rural: 28% and urban: 21.8%). However, no significant (*p* = 0.0898) association of allelic frequency was observed with the study site (Table 2). Most of the alleles were observed at a frequency of less than 10% (Figure 1), and more than half (n = 41, or 60.29%) of these alleles were common to the two study sites, including 18 of type K1, 14 of type MAD20, and 9 of type RO33. The zone-specific alleles were mostly found in rural areas: (10 K1 alleles, 10 MAD20 alleles, and 3 R033 alleles). For the *msp-2* gene (Table 2), a total of 46 genotypes were detected in rural areas (26 FC27 alleles: 200–760 bp and 20 3D7 alleles:120–500 bp), compared to 44 identified in urban areas, including 22 FC27 alleles (220–680 bp) and 22 3D7 alleles (120–700 bp). In total, 13% of the isolates were non-identified. The frequency of the FC27 allelic family was superior both in rural (64%) and urban (54.5%) areas compared to 3D7 allelic family (rural: 36% and urban: 45.5%). A significant (*p* = 0.0098) association of allelic frequency was observed with the study site (Table 2). The totals of the most frequent alleles in rural areas were FC27 440 bp, 460 bp, and 3D7 240 bp. In contrast, FC27 460 bp and 3D7 180 bp and 400 bp alleles were predominant in urban areas (Figure 2).

### 3.3. Allelic Frequency of P. falciparum msp-1 and msp-2 Genes According to Season, Age, and Clinical Status

The msp-1 and msp-2 allelic frequencies were also compared in this study. In rural areas, the K1 allelic family was predominant in the rainy season (43.2%), while the RO33 allelic family was the most abundant in the dry season (41.3%) (Appendix A). In urban areas, the K1 allelic family was predominant in both the rainy (42.6%) and dry (18.4%) seasons. There was a significant association between the allelic frequency and the season in rural areas (*p* = 0.0002) but not in urban areas (*p* = 0.0839). The K1 allelic family was predominant in all age groups both in rural and urban areas, and a significant association of the allelic frequency was observed with the age groups in rural areas (*p* < 0.0001) but not in urban area (0.5568). No significant association of the allelic frequency was found with the type of infection (microscopic and sub-microscopic) or with malaria clinical status (symptomatic and asymptomatic) in the two study sites. Among the sociodemographic parameters used for analyzing the *msp-2* allelic frequencies (Appendix A), only the age groups were significantly (*p* = 0.0113) associated with allelic frequency in rural areas. The 3D7-type alleles were predominant in children less than 5 years (52.2%) of age, while FC27-type alleles were found to be more abundant in participants aged 5 to 14 years (67%) and >15 years (64.4%).

### 3.4. Polyclonal Infections and Multiplicity of Infection According to Season, Age, and Clinical Status of the Participants

The characterizations of *msp-1* and *msp-2* genes showed totals of 1743 (879 for *msp-1* and 864 for *msp-2*) and 473 (262 for *msp-1* and 211 for *msp-2*) individual fragments in rural and urban areas, respectively. Overall, the *P. falciparum* multiplicity of infection (*msp-1*+*msp-2*) was significantly (*p* = 0.0006) higher in rural (2.9) areas compared to the urban areas (2.4). However, there was no significant (*p* = 0.1006) difference in polyclonal infection (PI) rate between the two areas (rural PI: 82.1% (316/385) vs. urban PI: 75.6% (102/135)). The proportion of polyclonal infection was significantly (*p* < 0.0001) higher in rainy season compared to the dry season, while this parameter was similar between the two seasons in the urban zone (Table 3). However, the rainy season was characterized by significantly (*p* < 0.0001) higher *P falciparum* MOI compared to the dry season both in rural and urban areas. The grouping of participants according to the type of infection (Table 3) in each study area showed a significantly (*p* < 0.05) lower rate of polyclonal infections in sub-microscopic infected (rural PI: 76.6%; urban PI: 70.5%) participants compared to their counterparts presenting a microscopic infection (rural PI: 90.3%; urban PI: 93.1%). The same trend was true when comparing *P. falciparum* MOI between participants presenting microscopic and sub-microscopic infection in the two areas. No association was observed when the proportions of polyclonal infections and the MOI were compared according to the age groups and the clinical status of the participants.

## 4. Discussion

Understanding the genetic diversity of *P. falciparum* in different geographical contexts is of crucial importance for developing new and effective control strategies [3]. To the best of our knowledge, no study assessing the genetic diversity and multiplicity of *P. falciparum* infection in rural areas in comparison with urban areas using both msp-1 and msp-2 gene markers has been conducted in the Republic of Congo since the implementation of malaria control in 2006.

The present work showed a high genetic diversity of *msp-1* and *msp-2* in the study sites, but this was significantly higher in rural areas compared to urban areas. Site-specific alleles were common in rural areas in comparison with the urban zone, as reported in Burkina Faso [17] and Gabon [18]. The disparity of polymorphism between the study areas might be due to some socioecological factors, including the diversity of malaria vectors in each locality, the behavior of the inhabitant in the malaria control (household cleaning and used of insecticide bed net), and climate change (humidity, precipitation). This high diversity found in rural areas suggests a high risk of being infected by distinct parasite strains and likely underlies a higher prevalence of *P. falciparum* infection in rural areas compared to urban areas [5]. The level of polymorphism found in urban areas in this present study is also consistent with the previous findings in the same area after introduction of ACTs and mass deployment of insecticide-treated nets in the country [19,20,21]. In contrast with the previous studies, this work was carried out during the COVID-19 outbreak, when malaria control measures were almost abandoned and the population was reluctant to admit patients to health centers for malaria treatment. This study showed a predominance of the K1/*msp-1* and FC27/*msp-2* allelic families both in rural and urban areas and was in line with previous reports in urban areas in the Republic of Congo [19,20,21].

A significant association of the *msp-1* allelic frequencies was observed with season and age groups in rural areas in this study but not in urban areas. In rural areas, the K1 allelic family and the RO33 allelic family were predominant in the rainy and dry seasons, respectively, while the urban areas were dominated by the K1 allelic family during the two seasons. The K1 allelic family was predominant in all age groups both in rural and urban areas. The *msp-2* allelic frequency was significantly associated with age group in rural areas, with 3D7-type alleles predominating in children less than 5 years of age and FC27 type alleles predominating in participants more than 5 years of age. Taken together, these findings suggest the capacity of *P. falciparum* to adapt itself to the changing environment and seasonality and to the level of immune response of the host. Overall, the genetic diversity of the three allelic families in these study areas was higher compared to what has been reported in several localities in Cameroon [22] and Gabon [18].

The MOI is an important parameter for the evaluation of level of malaria transmission and human immunity in a given locality [23]. In this study, the multiple infection rate and mean of MOI varied between study sites and were higher in rural areas than in urban areas. This high multiple infection rate and MOI suggest high malaria transmission, which is in line with the high malaria prevalence reported in rural areas [5]. This might be justified by the limitation of the control strategies used or their disproportional deployment on one hand and on other hand by the climatic and geographical conditions favoring the expansion of vectors responsible for the maintenance of malaria transmission in endemic areas [24,25].

In addition to the type of transmission area (rural versus urban), the season and the type of infection (microscopic or sub-microscopic) were also factors associated with the variation of the number of parasite genotypes in this study. The rate of polyclonal infection and *P. falciparum* MOI were significantly higher during the rainy season compared to the dry season. This result suggests an optimal malaria transmission likely coordinated by a high biodiversity of Anopheles mosquitoes during the rainy season, which is a conducive factor to malaria vector development [26]. A recent study conducted in Burkina Faso also reported that participants infected with *P. falciparum* harbored a higher number of parasite genotypes during the rainy season compared to the dry season [27]. Several studies carried out in a hospital setting in the Republic of Congo reported a high multiplicity of infection of *P. falciparum* in children with a microscopic infection compared to those with a sub-microscopic infection [20,21]. The same trend of results was found in this study, suggesting a multiplication of uncontrolled parasite strains in the population [28].

In contrast to our study in which the *P. falciparum* MOI was similar between the symptomatic and the asymptomatic patients, the findings of a longitudinal study reported a decrease of *P. falciparum* multiplicity infection when the clinical status of children moved from the asymptomatic to the symptomatic stage [29]. However, it is still difficult to determine whether the observed low multiplicity of infection in symptomatic individual is a cause or consequence of clinical symptoms. Although it is well documented that the age influences the multiplicity of infection [17,18], it was not the case in this study, as also shown previously in the Republic of Congo and Mali [21,30].

## 5. Conclusions

This study provides new data on the genetic diversity of *P. falciparum* isolates in rural and urban settings south of Brazzaville after the deployment of numerous malaria control measures in 2006. The findings revealed high allelic diversity and MOI of *P. falciparum,* reflecting high intensity of malaria transmission in rural areas. The rainy season and the presence of microscopic infection were associated with increases in diversity of *P. falciparum* isolates. Thus, these findings will contribute to strengthening malaria control measures, taking into account factors such as the seasonality and the level of malaria transmission in the areas.

## Figures and Tables

**Figure 1 pathogens-12-00742-f001:**
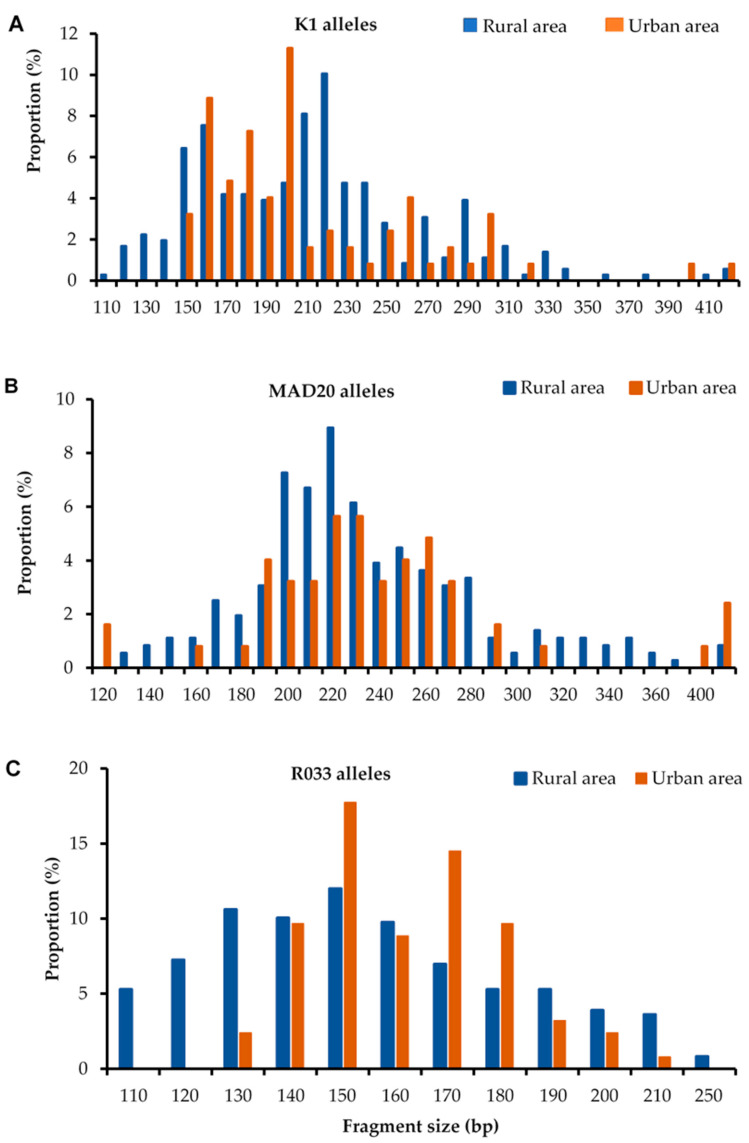
Distribution of msp-1 K1 (**A**), Mad20 (**B**), and Ro33 (**C**) alleles by area.

**Figure 2 pathogens-12-00742-f002:**
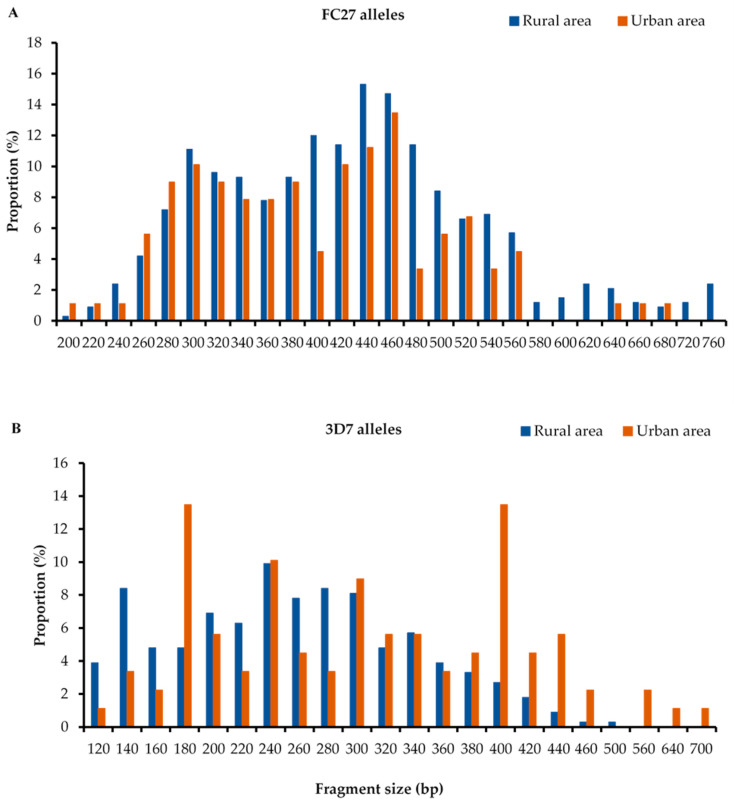
Distribution of msp2 FC27 (**A**) and 3D7 (**B**) alleles by area.

**Table 1 pathogens-12-00742-t001:** General characteristics of the study population.

Parameters	Rural Area (n = 397)	Urban Area (n = 151)	Total (n = 548)
**Season n (%)**			
Dry	197 (49.6)	75 (49.7)	272 (49.6)
Rainy	200 (50.4)	76 (50.3)	276 (50.4)
Sex ratio (male/female)	0.9 (195/202)	0.6 (59/92)	0.8 (255/294)
Age (years) ^a^	14 (1–88)	16 (3–81)	15 (1–88)
**Age group, years n (%)**			
<5	38 (9.6)	8 (5.3)	46 (8.4)
[5–15]	164 (41.9)	59 (39.1)	223 (40.7)
>15	195 (49.1)	84 (55.6)	279 (50.9)
**Parasitemia (para/µL) ^c^**	753 (575–985)	592 (290–1208)	724 (563–930)
**Type of infection n (%)**			
Microscopic	155 (39.1)	30 (19.9)	185 (33.7)
Sub-microscopic	242 (60.9)	121 (80.1)	363 (66.2)
**Clinical status**	**n1 = 155**	**n2 = 30**	
Asymptomatic	87 (56.1)	19 (63.3)	106 (57.3)
Symptomatic	68 (43.9)	11 (36.7)	79 (42.7)

a: median (range). c: geometric mean (95% confidence interval (95% CI)). n1: total number of participants in rural area. n2: total number of participants in urban area.

**Table 2 pathogens-12-00742-t002:** Frequency of *msp-1* and *msp-2* allelic families in rural and urban areas.

Allelic Family	Rural Area (n = 397)				Urban Area (n = 151)				p1	p2
	Prevalence n (%)	Frag Size (bp)	No. of Different Alleles	Allele Frequency n (%)	Prevalence N (%)	Frag Size (bp)	No. of Different Alleles	Allele Frequency n (%)		
** *msp-1* **										
K1	260 (65.5)	120–420	28	344 (39.0)	104 (68.9)	150–420	19	119 (45.4)	0.4537	0.0898
MAD20	220 (55.4)	130–410	26	245 (28.0)	49 (32.5)	120–410	16	57 (21.8)	<0.0001	
R033	288 (72.5)	110–250	12	290 (33.0)	86 (57.0)	130–210	9	86 (32.8)	0.0005	
Total			66	879			44	262		
** *msp-2* **										
FC27	291 (73.3)	200–760	26	554 (64.0)	57 (37.7)	220–680	22	115 (54.5)	<0.0001	0.0098
3D7	205 (51.6)	120–500	20	310 (36.0)	62 (41.1)	120–700	22	96 (45.5)	0.0269	
**Total**			46	864			44	211		

n: number of isolates. p1: comparison of prevalence to allelic families between rural and urban areas. p2: comparison of allelic frequency according to the study areas.

**Table 3 pathogens-12-00742-t003:** Polyclonal infections (PI) and multiplicity of *P. falciparum* infection according to season and age and clinical status of the participants.

Parameters	n1	Rural				n2	Urban			
		PI n (%)	P1	MOI	P2		PI n (%)	P3	MOI	P4
**Seasons**										
Dry	190	132 (69.5)	<0.0001	2.1	<0.0001	67	47 (70.1)	0.1468	1.9	<0.0001
Rainy	195	184 (94.4)		3.6		68	55 (80.9)		3.0	
**Age groups (years)**										
<5	38	27 (81.1)	0.1213	2.6	0.3174	6	3 (50)	0.0940	2.2	0.6028
[5–15]	157	129 (82.2)		2.9		54	39 (72.2)		2.6	
>15	191	160 (83.8)		3.0		75	60 (80.0)		2.4	
**Type of infection**										
Microscopic	154	139 (90.3)	0.0006	3.2	0.0003	29	27 (93.1)	0.0137	2.7	0.0003
Sub-microscopic	231	177 (76.6)		2.7		105	74 (70.5)		2.0	
**Clinical status**										
Asymptomatic	87	82 (94.3)	0.0569	3.4	0.0506	17	15 (88.2)	0.0173	3.0	0.6792
Symptomatic	67	57 (85.1)		2.9		11	11 (100)		3.4	

n1: total number of patients for each parameter in rural areas. n2: total number of patients for each parameter in urban areas. PI: polyclonal infection. MOI: multiplicity of infections.

## Data Availability

All data are fully available without restriction. Data are available from the FCRM Institutional Data Access. All request for data should be addressed to the Executive Director of FCRM, reachable at the following address: Prof. Francine Ntoumi, villa D6, Cité OMS Djoué, Brazzaville, République du Congo (Tel: +242 06 997 79 80, email: fntoumi@fcrm-congo.com).

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
