# Peer review of "Comparative study of Plasmodium falciparum msp-1 and msp-2 Genetic Diversity in Isolates from Rural and Urban Areas in the South of Brazzaville, Republic of Congo"

_pathogens, 2023, doi:10.3390/pathogens12050742_

Round 1

Reviewer 1 Report

The genetic diversity of Plasmodium falciparum infections in urban and rural areas of the Republic of Congo is assessed by the authors. Allele-specific PCR is used to analyze the pfmsp1 and pfmsp2 markers.

The study is interesting and has the potential to be published, but before further assessing the document, it is imperative to enhance the language quality and the clarity of several paragraphs, particularly in the presentation of results.

Many grammatical faults make it impossible to accurately judge the document's quality. Whenever the authors improve the quality of the English language, I would like to provide the document with a full assessment.

Author Response

Dear Reviewer,

Reviewer 2 Report

The authors genotyped msp-1 and msp-2 polymorphic genes in P. falciparum isolates collected from 548 participants in two villages situated in the south of the capital city and in an urban area in Brazzaville during rainy and dry seasons in 2021.

The background section is adequate. The methods are well described and are straightforward. The results are described clearly in detail, except for data presented in Table 1. As expected, Pf isolates in rural areas were characterized by a higher level of diversity and multiplicity of infection (MOI). More detailed discussion (an additional 1 paragraph) on genetic diversity and MOI, notably a comparison of results with previous studies performed in the Republic of Congo, can be added.  

English needs extensive corrections.

Note: My review does not include the authors’ supplementary data, which I was not able to download.

MAJOR COMMENTS:

Table 1:

The following data should be clearly presented, including the number and percentage (with the denominator of 548 for the total number of samples and/or the denominator of 397 for rural areas and 151 for the urban area):

1-the number of microscopy-positive, symptomatic cases,

2-the number of microscopy-positive, asymptomatic cases,

3-the number of microscopy-negative but PCR-positive cases, and

4-the number of microscopy-negative and PCR-negative cases.

Table 1, row “clinical status”: The legend says that there were 155 (total number) of participants in the rural areas and 30 (total number) participants in the urban area. 155+30= 185. This is confusing. It cannot be the total numbers because they do not add up to 548. Please clarify.

Table 1, line 189: median (range) instead of “min-max”; Age group, Total 164+59=223 (not “129”); 195+84=279 (not “114”); “b” is not indicated in the Table; “c” geometric mean (95% confidence interval [95% CI])

Table 1 (row “type of infection"), Results (lines 183-184): It is not clear how many of 548 participants were positive for malaria. The table seems to say that all 548 were positive for malaria (185 microscopy-positive + 363 microscopy-negative but PCR-positive). This important point should be clarified, both in the text (lines 183-184) and in the table.

Conclusion, lines 337-343: The authors summarize their findings. A more appropriate concluding statement should be added in line 343.

MINOR COMMENTS:

Line 3: The title must be written in correct English: in the south of Brazzaville, Republic of Congo

Line 7: Please write the country name in English (Republic of Congo)

Line 18: Plasmodium falciparum merozoite surface protein-1 (msp-1) and merozoite surface protein-2 (msp-2)

Lines 18-19, “from rural from rural”...has not been reported

Line 23: The genes coding for merozoite surface protein-1 and -2

Line 25: allelic families were predominant

Line 28: “the presence of microscopic infection...associated with an increase of MOI” The meaning is not clear.

Line 29: Republic (capital letter R)

Line 35: delete “(P. falciparum)”

Line 68: delete “either”

Line 104: ethylenediaminetetraacetic acid (EDTA)

Line 108: The “Lambarene method” is not clear. Is it a thick smear or thin smear?

Line 118: World Health Organization (WHO)

Line 131: amplicons...were

Line 132, “separated couple of primers”: different pairs of primers

Line 135: MgCl2 (in subscript)

Line 142: New England Biolabs

Line 146, “where”: were those with...

Line 152, “sub-microscopy infection”: submicroscopic infection

Lines 163-164: Kruskal-Wallis test

Line 165: analysis of variance (ANOVA)

Lines 175-177, “This section may be divided... can be drawn”: Please delete these lines.

Line 180: delete “in general”

Line 183: women were

Lines 183-184, “individuals with microscopic infection”: Individuals with positive microscopy...among whom (not “which”)

Line 186: asexual parasites/µL

Line 194: were found most frequently in isolates collected in rural areas

Line 196: the prevalence...was

Line 212, “22 3D7 alleles": please check the punctuation.

Line 213, “non-identified to any of the...family”: were non-identified. (period)

Line 220, Table 2: Title - Frequency of msp-1 and msp-2 allelic families; column title – no of different alleles; “msp-2” in row 8 should be placed and aligned in the first column.

Line 224: “msp1” in italics

Line 227: “msp2” in italics

Line 230, “The comparation...was also reported”: The msp-1 and msp-2 allelic frequencies were also compared in this study.

Line 233: both in rainy and dry seasons

Line 238: No (capital letter N) significant association

Line 255: significantly higher

Line 258: significantly lower rate

Line 262: when the proportions...were compared; clinical (small letter “c”) status

Line 266, Table 3: It is somewhat difficult to understand this table. It may become clearer if the column “n2” is separated from “n1”. The column “n1” can be left where it is. The column “n2” can be placed before “PI n (%)” of Urban area.

Line 273: in comparison with

Line 280: due to

Line 282: climate change

Line 283: distinct parasite strains

Line 284: likely underlies (instead of justify) or is likely associated with a higher prevalence

Line 287: mass deployment of insecticide treated nets

Line 287: In contrast to the previous studies

Line 289: population was

Line 298: The K1...both...areas

Line 302: to the changing environment

Line 304: compared to what has been reported

Line 307: in a given locality

Line 308: higher in rural area than in the urban area

Line 309: infection rate and MOI suggest

Line 313: vectors responsible for maintenance of malaria transmission

Line 315: in addition to the type of transmission area (rural versus urban)

Line 318-321, “This result suggests an optimal action...in tropical areas”: The meaning of the sentence is not clear. Please rewrite.

Line 327: was found (not “true”) in this study

Line 331: multiplicity of infection

Lines 331-332: clinical status of children changed from asymptomatic to symptomatic malaria

Line 334: that age influences multiplicity of infection

Line 335: Republic

Ref 4: Please complete this reference and add the web link. There is no title of the document (?).

Ref 18: Please use the same format throughout. Please see the article title of this reference, compared to other cited references.

extensive editing required

Author Response

Dear reviewer,

Round 2

Reviewer 1 Report

The authors have improved the quality of the language and have tried to correct the large number of errors present in the original manuscript. However, there are still some errors throughout the document, which have been highlighted in yellow in the attached pdf.

On the other hand, it is suggested to replace the first sentence of the abstract with the following sentence: "Polymorphisms in the genes encoding the merozoite surface proteins msp-1 and msp-2 are widely used markers to characterize the genetic diversity of Plasmodium falciparum. This study aimed to compare the genetic diversity of circulating parasite strains in rural and urban settings in the Republic of Congo after the introduction of artemisinin-based combination therapy (ACT) in 2006".

Author Response

Dear Reviewer, 
